# Attitude-Independent Route Tracking for Subsea Power Cables Using a Scalar Magnetometer under High Sea Conditions

**Guozhu Li [1], Xuxing Geng [1], Shangqing Liang [2,3,4], Yuanpeng Chen [1], Guangming Huang [1,\*], Gaoxiang Li [1], Xueting Zhang [3] and Guoqing Yang [2,3,4]**

1   Department of Physics, Central China Normal University, Wuhan 430079, China; lgz@mails.ccnu.edu.cn (G.L.); xxg@ccnu.edu.cn (X.G.); chen_yuanpeng_zstu@163.com (Y.C.); gaox@mail.ccnu.edu.cn (G.L.)
2   School of Electronics and Information, Hangzhou Dianzi University, Hangzhou 310018, China; liangshangqing@hdu.edu.cn (S.L.); gqyang@hdu.edu.cn (G.Y.)
3   Ocean Technology and Equipment Center, Hangzhou Dianzi University, Hangzhou 310018, China; zxt@hdu.edu.cn
4   Institute of Detection, Early Warning and Information Countermeasure, Hangzhou Dianzi University, Hangzhou 310018, China
\*   Correspondence: gmhuang@mail.ccnu.edu.cn

**Abstract:** To overcome the shortcoming wherein the accuracy of subsea cable detection can be affected by the determination of the bias vector, scale factors, and non-orthogonality corrections of the vector magnetometer, a real-time attitude-independent route tracking method for subsea power cables is investigated theoretically and experimentally by means of scalar magnetic field checking. The measurement of the magnetic field $B_c$ produced by the current in a cable is made immune to the influence of the platform attitude by extracting the component of $B_c$ along the geomagnetic field using a high-bandwidth self-oscillating optically pumped magnetometer. The self-oscillating frequency is proved to be independent of the attitude of the magnetometer with the theoretical model. Experiments are carried out to test the attitude-independent performance, and the effectiveness of route tracking is verified by the results of the sea experiment. The proposed method will effectively improve the ability to locate subsea cables under high sea conditions.

**Keywords:** scalar magnetometer; subsea power cables; attitude-independent detection; route tracking; high sea condition

## 1. Introduction

Subsea cables are the main carrier for integrating energy into the onshore grid [1]. However, these cables face potential risks of damage from both natural factors and human activities. Human activities are the primary cause of cable faults, such as fishing and anchoring [2,3]. Matrix-type subsea power cable protectors assembled with reinforced concrete blocks have been tested to protect cables from anchoring [4]. Cables can also be damaged by natural causes, including ocean currents [5] and earthquakes [6,7]. When a cable is damaged, it is necessary to locate the fault as soon as possible to reduce economic losses [8]. Detailed subsea cable route maps serve as valuable references for locating cable faults, enabling significant time savings. Consequently, the detection of subsea cables has become a top priority, especially cable route tracking.

The methods of subsea cable detection mainly include acoustics, optics, electricity, and magnetism. Probe devices are usually integrated into AUVs or ROVs [9,10]. Sonar is the main equipment used for acoustic detection. High-resolution sonar instruments, such as multibeam sonar [10], synthetic aperture sonar [11], and side-scan sonar [12], are needed for subsea cable-like object detection. In order to widen the frequency band, biosonar is used for cable detection [13]. Acoustic detection is mainly used for cables that are exposed to

the seabed or buried shallowly in the seabed [14]. Vision technology can provide real-time observation for optical detection. In order to solve the problem wherein vision technology has no effect in a turbid sea environment, some methods, including those integrating image enhancement [15], image segmentation [16], and particle filters [17], have been proposed. In addition, the high-precision recognition of curved cables has been realized by means of edge detection [18,19] and the random sample consensus (RANSAC) algorithm [19]. Acoustic and optical methods are only applicable for detecting subsea cables that are exposed on the seabed or shallowly buried. When the cables are deeply buried beneath the seabed, both methods become ineffective. Electrical methods are mainly used in cable fault detection. Time–frequency domain reflectometry is a traditional method [20]. To solve the difficulty of high-impedance fault location, a deep learning algorithm [21] and a novel method by analyzing the variation difference in the equivalent current in the Laplace transform domain [22] have been proposed. When using electrical methods to locate the fault points of the subsea cables, precise cable route information is essential. Otherwise, it is challenging to accurately locate the positions of the fault points.

Magnetic detection is a popular method because cables do not need to be seen visually. Vector magnetic field sensors, including fluxgate [23,24] and multi-probe [25], are usually applied to cable detection. There are some commercial devices, such as the TSS 350 and MAG-032C. The disadvantage of these magnetic field sensors is that the detection results are affected by the determination of the magnetometer bias vector, scale factors, and non-orthogonality corrections [26]. To correct these errors, some calibration methods have been proposed, but changes in magnetic fields cannot be completely corrected. Figure 10 of Ref. [26] shows that, after interference component suppression, the respective magnetic field measurement errors in each direction under the background geomagnetic field are 163.07 nT (2.1%), 328.1 nT (4.28%), and 205.1 nT (2.53%). Ref. [27] used attitude information to calibrate vector magnetic field sensors and achieved a simulation error of several nT and a measurement error less than 30 nT under the background geomagnetic field. And these methods need algorithms to weaken the influence of the sensor attitude; thus, computational requirements are increased, which affects real-time performance. In refs. [23,24], the authors achieved very good results in automatic cable tracking with accuracies of <2.5 m and <1 m, shown in their plots. These papers share a common analytical method, which is to conduct analysis under the condition of ignoring the changes in platform attitude. In the first paragraph of Section II. B of Ref. [23], a theoretical model was proposed with the assumption that the roll and pitch motions of the autonomous underwater vehicles (AUV) are ignored. Similarly, in the first paragraph of Section IV of Ref. [24], the assumption was made that the AUV does not pitch or roll. On the ocean surface, the attitude of an AUV or remotely operated vehicle (ROV) is constantly changing, especially in high sea conditions. This causes significant measurement errors for the vector magnetometer, thereby affecting the positioning accuracy of the subsea cables.

The self-oscillating optically pumped magnetometer (OPM) is a high-sensitivity quantum scalar sensor that is widely used in aeromagnetic survey, marine monitoring, geological exploration, earthquake prediction, medical and health systems, and other fields. The optical properties of the OPM contribute to maintaining measurement accuracy with only a weak influence from the attitude of the magnetometer. This paper explores the utility of the OPM for cable positioning in high sea conditions, employing a scalar magnetic field checking method. In contrast, the proton magnetometer, despite being a scalar sensor, exhibits a response too slow to detect the 50 Hz signal. To achieve real-time tracking of subsea cables in challenging high sea conditions, the self-oscillating OPM becomes indispensable. This paper demonstrates that the measurement of the magnetic field $B_c$ generated by the current in a cable remains immune to platform attitude influences. This is achieved by extracting the $B_c$ component along the geomagnetic field using a high-bandwidth self-oscillating optically pumped magnetometer. In a laboratory experiment, the magnetometer tracks the cable at various postures to validate the attitude-independent performance of the OPM. In

a sea experiment, this paper reports the discovery of fifty-one points of subsea cables and the successful tracking of nine cable routes.

## 2. Attitude Independence for a Self-Oscillation OPM

OPMs are mainly applied for weak magnetic field detection. They are scalar sensors using the quantum effect wherein the motion state of alkali metal atoms can be affected by laser fields and magnetic fields [28]. The properties of the OPM are as follows: the magnetic field resolution is 3 pT, the dynamic range is 15,000 to 105,000 nT [29], and the bandwidth is limited by the frequency counter, which has a sampling rate of 250 Hz. The self-oscillation OPM offers distinct and substantial benefits in the measurement of magnetic fields due to the principles of both optical pumping and self-oscillation. These advantages can be found on pages VIII to XI of Ref. [29]. Due to the properties of the self-oscillation OPM, the measurement accuracy is only weakly influenced by the attitude of the magnetometer. In addition, real-time performance is also necessary for detecting subsea cables. The response of the self-oscillation OPM is extremely fast, such that the signal of the 50 Hz magnetic field can be detected without any appreciable lag in response.

To analyze the influence of the angle $\varphi$ between the magnetic field $B_0$ to be measured and the direction of light propagation, the OPM model geometry is established as shown in Figure 1. An optically pumped atomic magnetometer consists of three main processes: state preparation, magnetic field–atom interaction, and optical detection [30]. In the process of state preparation, the depopulation process of the pump lamp and the repopulation process of spontaneous emission cause the atoms in ground state to establish orientation polarization [30]. In the processes of magnetic field–atom interaction, the magnetic field $B_0$ causes the magnetic sublevels to produce linear Zeeman splitting, which generates the Larmor precession frequency. The weak-radiofrequency magnetic field $B_{rf}(t) = B_{rf} \cos\left[\omega_{rf} t\right]$ is parallel to the propagation direction of light. When the frequency of $B_{rf}$ is equal to the Larmor precession frequency, the magnetic resonance phenomenon occurs, and the oriented polarization established during the processes of quantum state preparation is changed. In the processes of optical detection, the probe light detects the change in polarization due to the magnetic resonance.

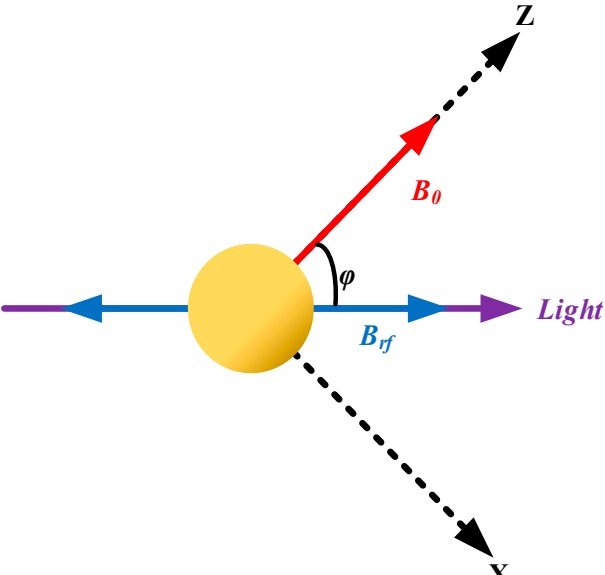

**Figure 1.** An OPM in the $B_{rf}$ and *light* geometry [30], where $B_0$ is taken as the Z axis and the vertical direction is taken as the X axis.

According to the analysis of the OPM, the Hamiltonian of the system is denoted

$$\mathcal{H} = \Delta F_0 + \Omega_{rf} F_{-1} - \Omega_{rf} F_1, \tag{1}$$

where the detuning $\Delta$ and Rabi frequency $\Omega_{rf}$ are $\omega_0 - \omega_{rf}$ and $\frac{\mu_B g_F B_{rf} \sin[\varphi]}{\sqrt{2}}$, respectively. The physical constants $\mu_B$ and $g_F$ are the Bohr magneton and the Landé factor. The operator $F_j (j = 0, \pm 1)$ is the angular momentum operator in the covariant spherical harmonic representation [31]. The dynamic evolution of the system satisfies the main equation $\dot{\rho} = -i[\mathcal{H}, \rho] + \mathcal{L}\rho$, where $\rho$ is the density matrix of the system and $\mathcal{L}\rho$ is the relaxation of the system [32]. Next, the density matrix $\rho$ is expanded into $\rho = \sum\limits_{k=0}^{2F} \sum\limits_{q=-k}^{k} m_{k,q} T_q^k$ using the irreducible tensor $T_q^k$, where $m_{k,q}$ is the k-order atomic multipole moment. Then, the signal of the OPM is obtained as

$$S(\varphi, \omega) = C_0 \sqrt{C_1^2 + C_2^2} \sin\left[\omega t + \arctan\left[\frac{C_1}{C_2}\right]\right], \tag{2}$$

where $C_1 = \frac{\sqrt{2} m_{0,0} \cos[\varphi] \Delta \Omega}{\Delta^2 + \Gamma^2}$ and $C_2 = \frac{\sqrt{2} m_{0,0} \cos[\varphi] \Gamma \Omega}{\Delta^2 + \Gamma^2}$. The parameters $\Gamma$ and $m_{0,0}$ are the relaxation and injection process of the system, respectively. The constant $C_0$ is related to parameters of the system such as the gain factor, transconductance amplification factor, and optical power [33]. The relationship between the signal amplitude of the OPM and the angle is shown in Figure 2, where the maximum SNR can be obtained at the angle of 45°. When an OPM is used for subsea cable detection, the equipment should be installed at a 45° angle to the geomagnetic field. According to stability standards established by the International Maritime Organization (IMO), the maximum swing angle of the ship should generally be less than 20° so that the amplitude of the signal is large enough to be collected.

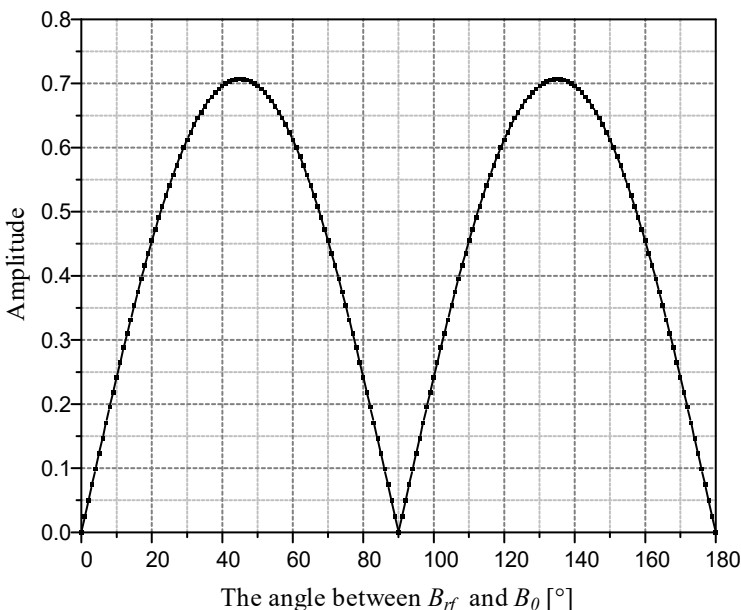

**Figure 2.** The curve of the output signal amplitude of the optically pumped magnetometer.

Through theoretical analysis, the self-oscillation OPM has real-time attitude-independent properties. When the OPM is used for subsea cables under the geomagnetic field, since it is a scalar sensor, the component of $B_c$ along the geomagnetic field will be detected, which will be used to locate subsea cables. The real-time attitude-independent properties of the OPM allow this method to be used in high sea conditions.

### 3. Positioning Method with the OPM for Subsea Cables

As the OPM functions as a scalar sensor, the analysis of the subsea power cable positioning method primarily involves extracting the component of $B_c$ along the geomagnetic field. This approach enables the scalar magnetometer to discern the cable's direction, angle, distance, and other relevant information. The real-time attitude-independent properties of the OPM contribute to obtaining this information with greater precision. The conventional technique for cable positioning often involves identifying peak or valley values in the magnetic field signal. In this paper, the approach of identifying the valley value is adopted.

According to the Biot–Savart law, the magnetic field $B$ of a long straight cable can be calculated by

$$B = \frac{\mu_0 I}{2\pi d}, \tag{3}$$

where the constant $\mu_0$ is the vacuum permeability value of $4\pi \times 10^{-7}$ H/m, $I$ is the current carried in the cable, and $d$ is the distance between the calculation point and the cable. The geomagnetic field and the magnetic field of the cable can be added together by the law of cosines. The equation is given by

$$B = \sqrt{B_g^2 + B_c^2 - 2B_g B_c \cos[\pi - \theta]}, \tag{4}$$

where $B_g$ is the geomagnetic field, $B_c$ is the magnetic field of the cable, and $\theta$ is the angle between both of them. Equation (4) can be converted to

$$B = B_g \sqrt{1 + \left(\frac{B_c}{B_g}\right)^2 + 2\frac{B_c}{B_g}\cos[\theta]}. \tag{5}$$

Equation (5) can be expressed with $\frac{B_c}{B_g}$ transformed to the variable x and then expanded as a Taylor series around x = 0, which is given by

$$f(x) = B_g\left(1 + \frac{\cos[\theta]}{1!}x + \frac{\sin^2[\theta]}{2!}x^2 + \frac{-3\sin^2[\theta]\cos[\theta]}{3!}x^3 + \dots + \frac{f^{(n)}(0)}{n!}x^n + o[x^n]\right) \tag{6}$$

Since $B_g$ is about 50,000 nT and $B_c$ is about a few hundred nT, the higher-order terms of $\frac{B_c}{B_g}$ can be neglected. The formula can be simplified to Equation (7) [30].

$$B = B_g + B_c \cos[\theta]. \tag{7}$$

The conclusion indicates that the magnetic field of the cable, measured through scalar checking, is approximately equal to the component of the cable's magnetic field along the geomagnetic field.

In order to analyze the component, a magnetic field model of the subsea power cable under the background geomagnetic field was established, as shown in Figure 3. In the model, the parameter $\alpha$ is the angle between the subsea power cable and the projection of the geomagnetic field on the sea level. The parameter $h$ is the height between the subsea power cable and the sea level. The parameter $\beta$ is the magnetic dip. The vector $\overrightarrow{OP}$ is the unit geomagnetic vector. The point $Q$ is a detection point. The vector $\overrightarrow{QR}$ is the magnetic field vector of the point $Q$.

A Cartesian coordinate system was established with the cable as the *x*-axis. The coordinates of $Q$ were set to $(0, y, h)$ and those of $R$ were set to $(0, y + a, z)$. The magnetic field $B_c$ of $Q$ is given by

$$B_c = \frac{\mu_0 I}{2\pi \sqrt{y^2 + h^2}}. \tag{8}$$

The angle $\theta$ between the vectors $\overrightarrow{OP} = (\cos[\beta] \cdot \cos[\alpha], \cos[\beta] \cdot \sin[\alpha], sin[\beta])$ and $\overrightarrow{QR} = (0, a, -\frac{ya}{h})$ is given by

$$\cos[\theta] = \frac{\overrightarrow{OP} \cdot \overrightarrow{QR}}{\left|\overrightarrow{OP}\right|\left|\overrightarrow{QR}\right|} = \frac{h \cos[\beta] \sin[\alpha] - y \sin[\beta]}{\sqrt{y^2 + h^2}}. \tag{9}$$

Finally, the component of $B_c$ along the geomagnetic field can be calculated by

$$B = B_c|\cos[\theta]| = \frac{\mu_0 I_0}{2\pi} \left| \frac{h \cos[\beta] \sin[\alpha] - y \sin[\beta]}{y^2 + h^2} \right|. \tag{10}$$

From Equation (10), there is a least value of 0 nT when

$$y_0 = \frac{h \sin[\alpha]}{\tan[\beta]}. \tag{11}$$

By calculating the first derivative of Equation (10), there are two peak values, $B_1$ and $B_2$, when

$$y_1 = \frac{h \sin[\alpha]}{\tan[\beta]} - \sqrt{\left(\frac{h \sin[\alpha]}{\tan[\beta]}\right)^2 + h^2}, \tag{12}$$

$$y_2 = \frac{h \sin[\alpha]}{\tan[\beta]} + \sqrt{\left(\frac{h \sin[\alpha]}{\tan[\beta]}\right)^2 + h^2}. \tag{13}$$

By Equations (11)–(13), the distance between $y_0$ and $y_1$ is equal to the distance between $y_0$ and $y_2$, which is shown as

$$d = \sqrt{\left(\frac{h \sin[\alpha]}{\tan[\beta]}\right)^2 + h^2}. \tag{14}$$

Additionally, the ratio of the magnetic field $B_1$ at $y_1$ to the magnetic field $B_2$ at $y_2$ is equal to the ratio of $y_2$ to $y_1$, which is shown as

$$p = \left(\frac{\sin[\alpha]}{\tan[\beta]} + \sqrt{\left(\frac{\sin[\alpha]}{\tan[\beta]}\right)^2 + 1}\right)^2. \tag{15}$$

During the experiment, $d$ can be measured using positioning equipment, $p$ can be measured using the OPM, and $\beta$ can be obtained by searching available information. By Equations (14) and (15), $h$ and $\sin \alpha$ can be derived as

$$h = \frac{2d}{p+1} \sqrt{p}, \tag{16}$$

$$\sin[\alpha] = \frac{p-1}{2\sqrt{p}}. \tag{17}$$

In our experimental area, it was found that the magnetic dip is 45 degrees and $I_0$ is a constant. Consequently, the magnetic field is related to $h$ and $\alpha$. Subsequently, we conducted a re-analysis to understand the influence of these two parameters on the calculation results. To demonstrate the impact of parameter $\alpha$ on the calculations, the value of $\mu_0 I_0$ was set to $2\pi$. In Figure 4, the graphical representation illustrates the effect of $\alpha$ on the magnetic field of the cable. As $\alpha$ increases, the position of the least value shifts. The valley value positions are noted at 0 m, 0.707 m, and 1 m. The offset distance can be calculated by Equation (11).

When *h* increases, the value of *d* becomes larger, but the shape of the magnetic field curve does not change.

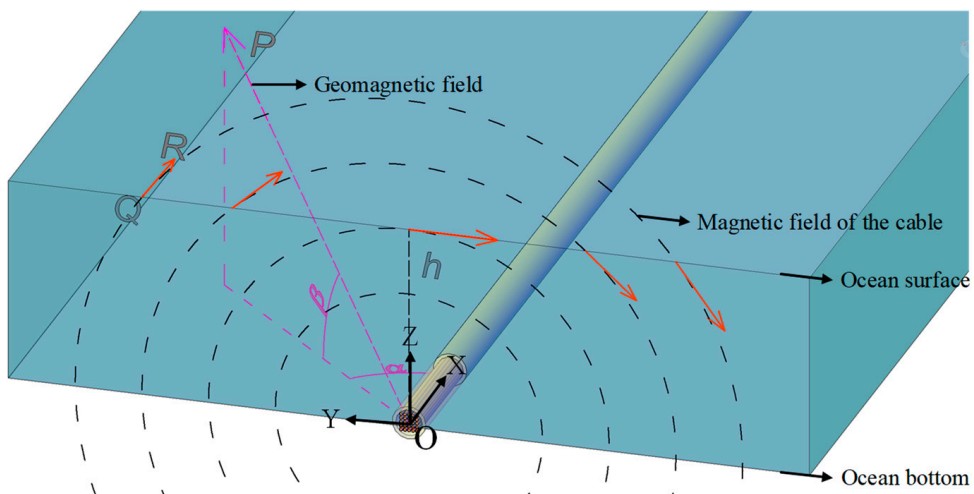

**Figure 3.** Magnetic field model of a subsea power cable under the background geomagnetic field. *h*: the height between the cable and the sea level; $\alpha$: the angle between the subsea power cable and the projection of the geomagnetic field; $\beta$: magnetic dip; $\overrightarrow{QR}$: the magnetic field vector of the cable; $\overrightarrow{OP}$: the unit geomagnetic vector.

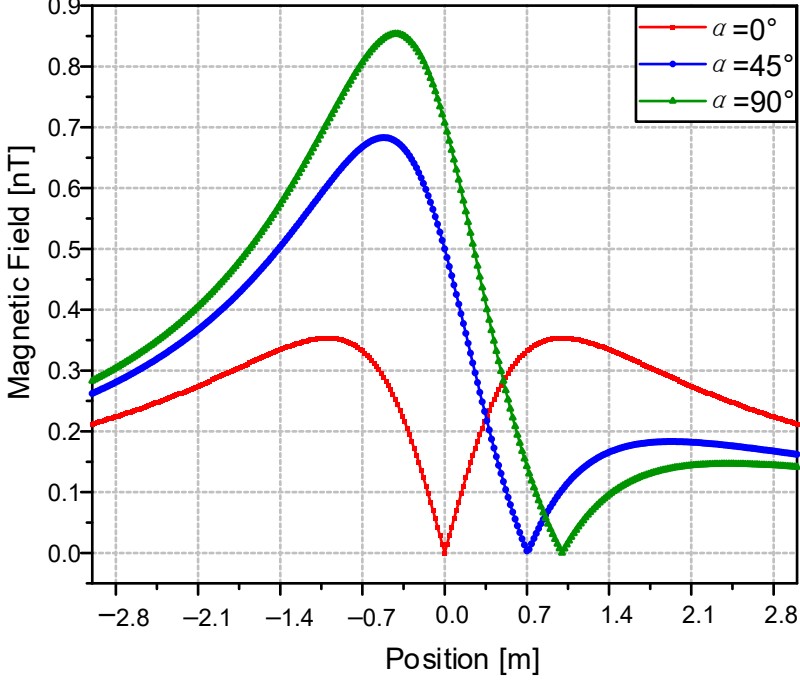

**Figure 4.** The magnetic field curves at $\alpha = 0°, 45°, 90°$; $h = 1$ m. The current $I_0$ and magnetic dip $\beta$ are both constants.

Hence, by utilizing GPS devices for location information and employing the OPM to capture the magnetic field of the subsea cable, it becomes feasible to gather data encompassing the cable's position, depth, and orientation.

The aforementioned concept introduces a theoretical framework for subsea cable positioning utilizing scalar magnetic field checking. To verify the attitude-independent feature of the OPM within this positioning approach, a series of experiments were conducted, including both indoor and simulation experiments.

## 4. Laboratory Experiment, Simulation Experiment, and Results Discussion

In order to validate the attitude-independent performance of the OPM, an indoor experiment was conducted. The OPM was positioned across angles ranging from −30 to +30 degrees. As for the positioning of multiple cables, due to the complexity of creating an experimental setup for multiple cables, a finite element simulation [34,35] was employed to simulate the magnetic field generated by multiple cables.

### 4.1. Laboratory Experiment

To examine the impact of the OPM's attitude on cable positioning results, the magnetometer was fixed on a non-magnetic rotation stage, as shown in Figure 5. For this indoor experiment, a cable was positioned on the floor, carrying a 250 mA (RMS) current at a frequency of 27 Hz. This frequency was chosen to facilitate data collection, with the magnetic field signal at 27 Hz extracted through FFT calculations. The vertical height between the cable and magnetometer was about 1.2 m, as shown in Figure 6. The magnetic dip was about 45 degrees. The angle between the cable and the projection of the geomagnetic field on the horizontal plane was about 20 degrees. The OPM crossed the cable vertically at angles of −30 degrees, −10 degrees, 0 degrees, 10 degrees, and 30 degrees and traveled about 4 m.

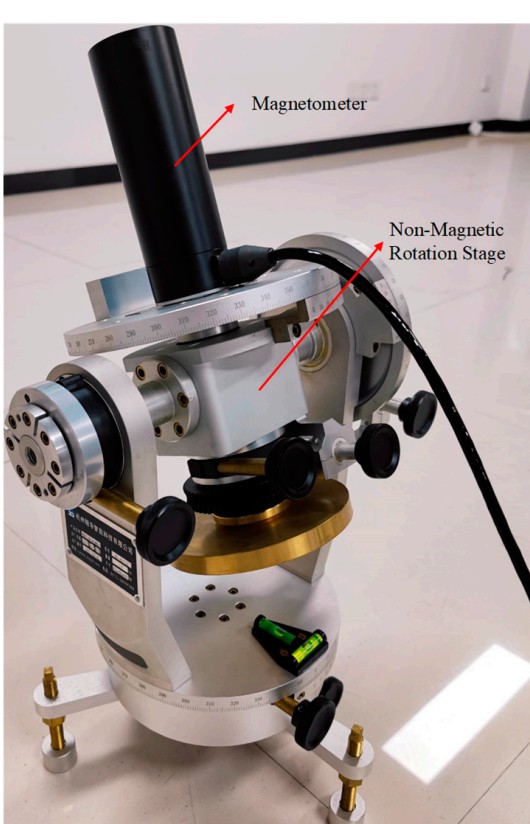

**Figure 5.** Attitude-independent performance verification platform.

The results are shown in Figure 7. Firstly, the maximum error was 4 nT at the position of −100 cm. This error may have been caused by the influence of other magnetic fields. On the ocean, the power of other magnetic fields is much lower, so the error can be ignored. Secondly, the position was recorded at 40 cm, while according to Equation (10), the calculated position was 41 cm. This discrepancy is attributed to the movement step length being set at 10 cm. Thirdly, the magnetic field curve is not symmetrical. The reason for this is that the direction of the cable was not parallel to the projection of the geomagnetic field. This result is consistent with the theoretical analysis. The cable positioning results are only

weakly influenced by the attitude of the OPM, so the attitude-independent performance of the OPM with scalar magnetic field checking is verified.

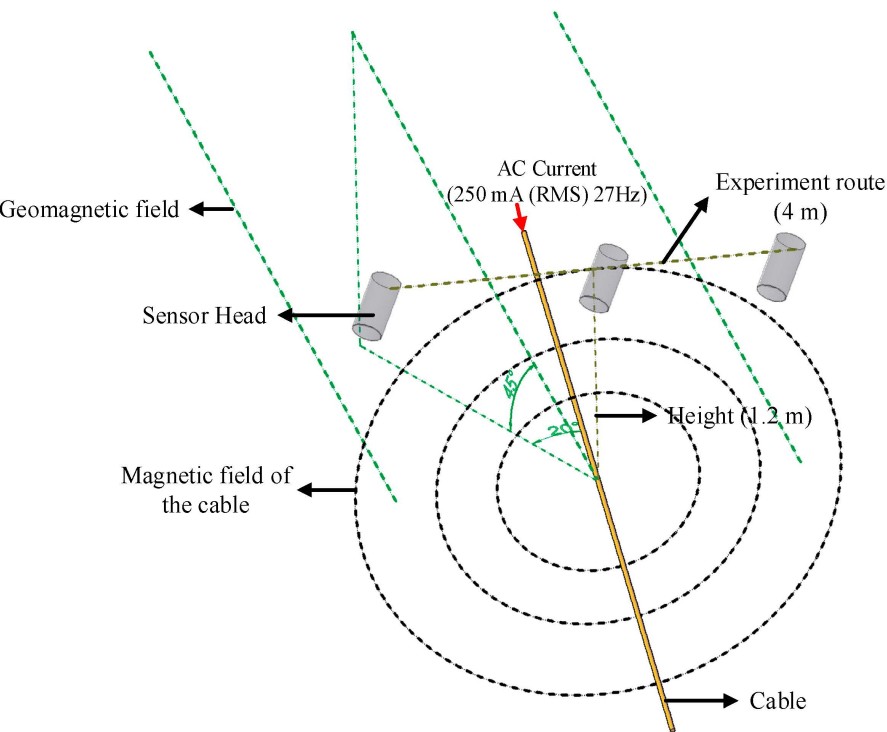

**Figure 6.** Indoor experiment diagram.

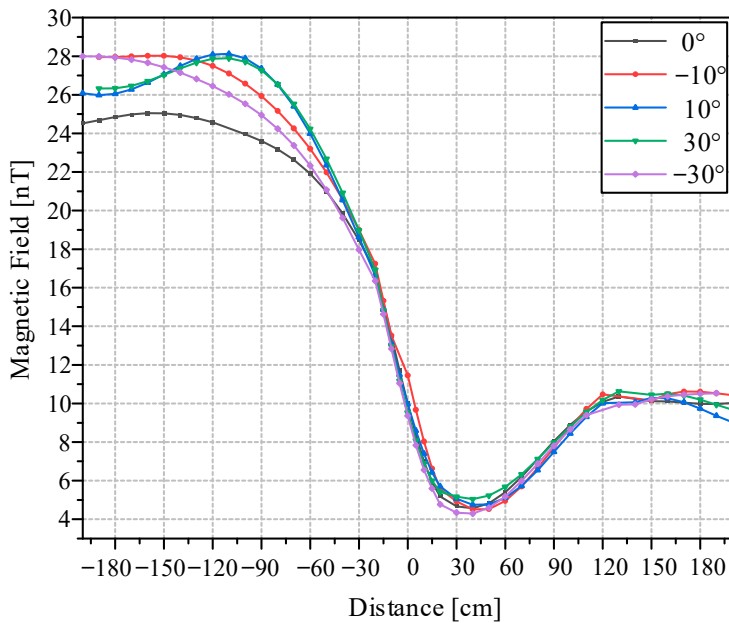

**Figure 7.** Results of the cable magnetic field under different magnetometer attitudes.

### 4.2. Simulation Experiment

The above is the experimental result for one cable. Usually, three cables are used as a set of transmission lines. To model the magnetic field of multiple cables, a finite element simulation was employed for comprehensive calculations. According to the map, there are three arrangements of three-phase subsea cables, amounting to a total of nine, and the distance between them is shown in Figure 8. Positioned above these cables is a 1 km long

and 20 m high observation line. The currents in the nine subsea cables are 21 A, 35 A, 21 A, 64 A, 42 A, 57 A, 42 A, 28 A, and 42 A, with a frequency of 50 Hz. The magnetic field was analyzed within 40 ms with a time step of 0.5 ms. According to the geometric relationship between the geomagnetic field and the cable magnetic field, the component of $B_C$ along the geomagnetic field was extracted.

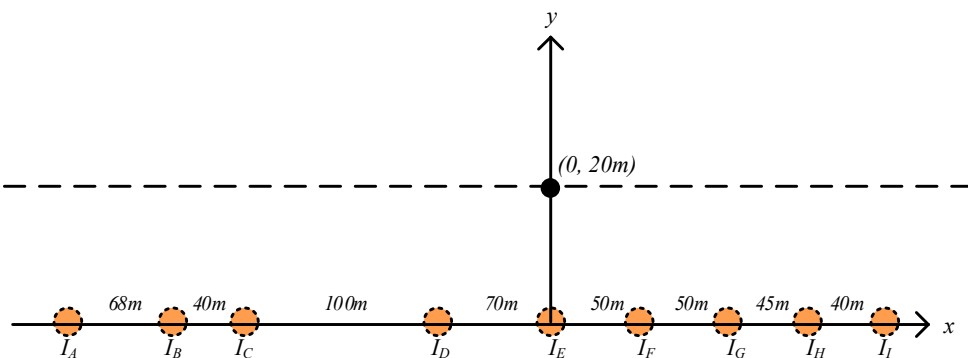

**Figure 8.** Simulation model of multiple cables.

The simulation outcomes are depicted in Figure 9. Each colored curve corresponds to the magnetic field fluctuation at a specific point in time. Evident within the curves are valley signals positioned above each cable, serving as indicators for cable localization. Notably, the applicability of the scalar magnetic field analysis method extends to scenarios involving multiple cables. In comparison to Figure 4, the peaks on either side of the cables overlap due to the close proximity of the cables. The simulation results' envelope can be utilized for comparison with magnetic field data (as illustrated in Figure 10) acquired during sea experiments.

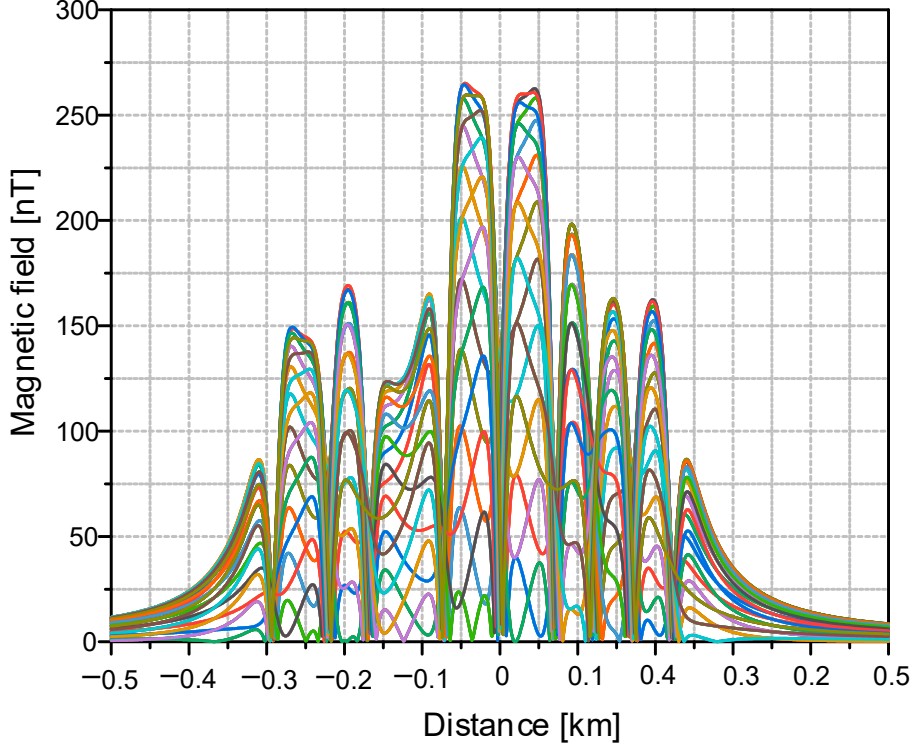

**Figure 9.** The simulation results of the magnetic field for multiple cables (different color lines represent different phases).

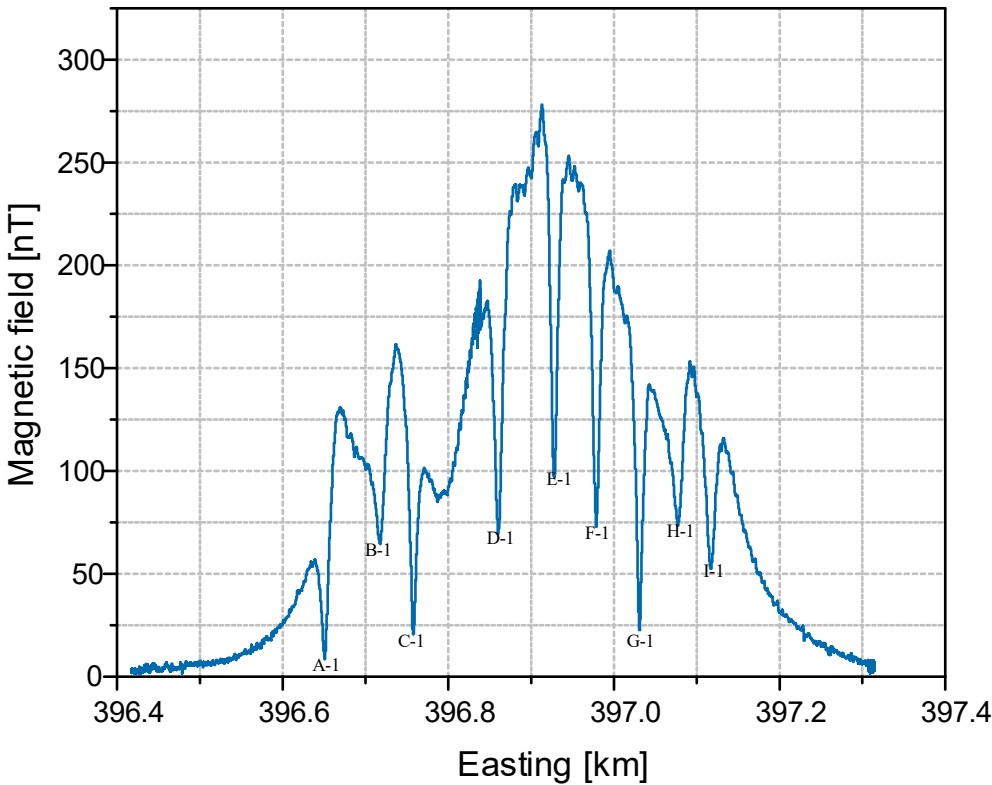

**Figure 10.** The experiment results of the magnetic field for multiple cables (9 positioning points of 9 cables).

To validate the accuracy of the simulation results, theoretical calculations were conducted for the magnetic field distribution of the nine subsea cables at sea level. According to Equation (8), the magnetic field $B_{C_A}$ in the horizontal plane at a depth of 20 m for subsea cable A is

$$B_{C_A} = \frac{\mu_0 I_A}{2\pi\sqrt{(278+x)^2 + 20^2}}, \tag{18}$$

where $I_A$ is the current carried in cable A, and x is the positional coordinate at the height of 20 m above sea level. According to Equations (9) and (10), where the magnetic dip $\beta$ is 45° and the angle $\alpha$ between the subsea cable and the projection of the geomagnetic field on the sea level is 0°, the magnetic field component $B'_{C_A}$ of the cable A along the geomagnetic field is

$$B'_{C_A} = B_{C_A} \cdot \cos[\theta_A] = \frac{\mu_0 I_A}{2\pi\sqrt{(278+x)^2 + 20^2}} \cdot \frac{\sqrt{2}(x+278)}{2\sqrt{(278+x)^2 + 20^2}}. \tag{19}$$

The current of subsea cable A is given by $I_A = 21\cos(2\pi \cdot 50t + 0°)$. Because the time variable $t$ can be expressed in terms of $x$ and the sailing speed, which was settled as 1 m/s, and calculation starts from the coordinates $(-500\text{ m}, 20\text{ m})$, the current of subsea cable

A can be expressed as $I_A = 21\cos[2\pi \cdot 50(x+500)+0°]$. So, the component $B'_C$ of the superposed magnetic field of the nine subsea cables along the geomagnetic field is

$$
\begin{aligned}
B'_C = {} & \frac{\mu_0 21\cos[2\pi\cdot50(x+500)+0°]}{2\pi\sqrt{(x+278)^2+20^2}} \cdot \frac{\sqrt{2}(x+278)}{2\sqrt{(x+278)^2+20^2}} \\
& + \frac{\mu_0 35\cos[2\pi\cdot50(x+500)+120°]}{2\pi\sqrt{(x+210)^2+20^2}} \cdot \frac{\sqrt{2}(x+210)}{2\sqrt{(x+210)^2+20^2}} \\
& + \frac{\mu_0 21\cos[2\pi\cdot50(x+500)+240°]}{2\pi\sqrt{(x+170)^2+20^2}} \cdot \frac{\sqrt{2}(x+170)}{2\sqrt{(x+170)^2+20^2}} \\
& + \frac{\mu_0 64\cos[2\pi\cdot50(x+500)+0°]}{2\pi\sqrt{(x+70)^2+20^2}} \cdot \frac{\sqrt{2}(x+70)}{2\sqrt{(x+70)^2+20^2}} \\
& + \frac{\mu_0 42\cos[2\pi\cdot50(x+500)+120°]}{2\pi\sqrt{(x+0)^2+20^2}} \cdot \frac{\sqrt{2}(x+0)}{2\sqrt{(x+0)^2+20^2}} \\
& + \frac{\mu_0 57\cos[2\pi\cdot50(x+500)+240°]}{2\pi\sqrt{(x-50)^2+20^2}} \cdot \frac{\sqrt{2}(x-50)}{2\sqrt{(x-50)^2+20^2}} \\
& + \frac{\mu_0 42\cos[2\pi\cdot50(x+500)+0°]}{2\pi\sqrt{(x-100)^2+20^2}} \cdot \frac{\sqrt{2}(x-100)}{2\sqrt{(x-100)^2+20^2}} \\
& + \frac{\mu_0 28\cos[2\pi\cdot50(x+500)+120°]}{2\pi\sqrt{(x-145)^2+20^2}} \cdot \frac{\sqrt{2}(x-145)}{2\sqrt{(x-145)^2+20^2}} \\
& + \frac{\mu_0 42\cos[2\pi\cdot50(x+500)+240°]}{2\pi\sqrt{(x-185)^2+20^2}} \cdot \frac{\sqrt{2}(x-185)}{2\sqrt{(x-185)^2+20^2}}
\end{aligned}
\tag{20}
$$

Curve plotting for Equation (20) was performed, as shown in Figure 11. This result matches well with the simulation results in Figure 9 and the sea experiment results in Figure 10.

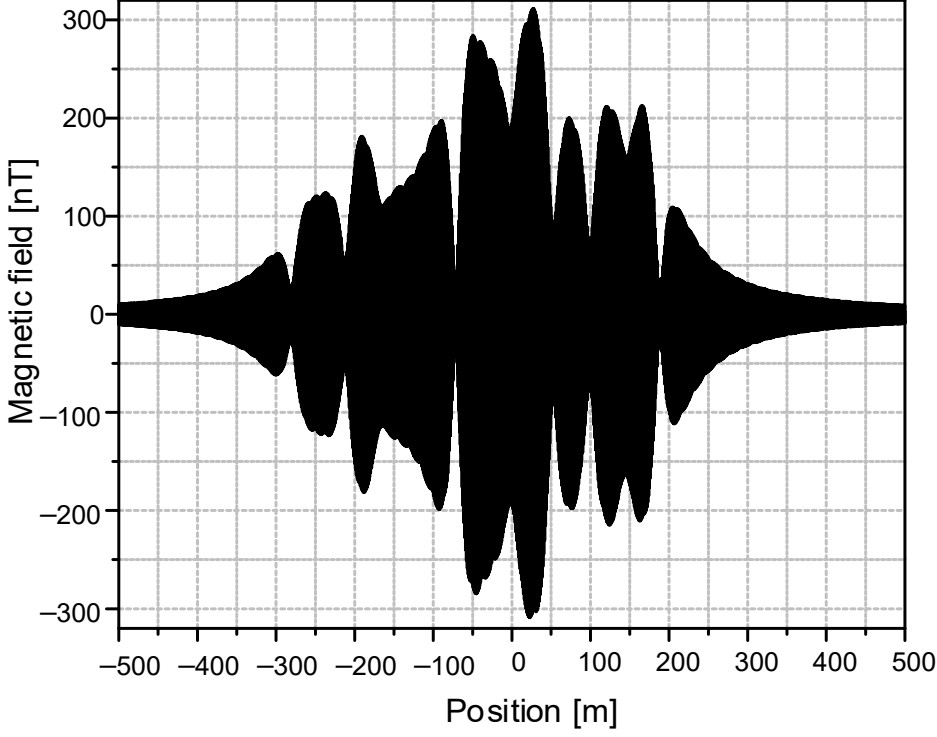

**Figure 11.** The calculation results of the magnetic field for multiple subsea cables.

## 5. Sea Experiment and Results Discussion

To validate the viability of subsea cable route tracking, a sea experiment was conducted. An optically pumped magnetometer was positioned 2 m from the bow of the vessel, as illustrated in Figure 12. The magnetic dip was approximately 45 degrees. According to the map, the experimental region hosted nine subsea power cables, oriented in a north–south

direction. These cables, marked from A to I, were situated at an approximate depth of 20 m. The primary objective of this experiment was to detect cables G to I. To achieve this, the ship maneuvered around these three cables, while the status of the remaining six cables regarding possible crossings remained uncertain. Throughout this operation, the area was traversed approximately 7 times in an east–west direction, encompassing a duration of about 1 h. The length of the cables under detection spanned roughly 2 km.

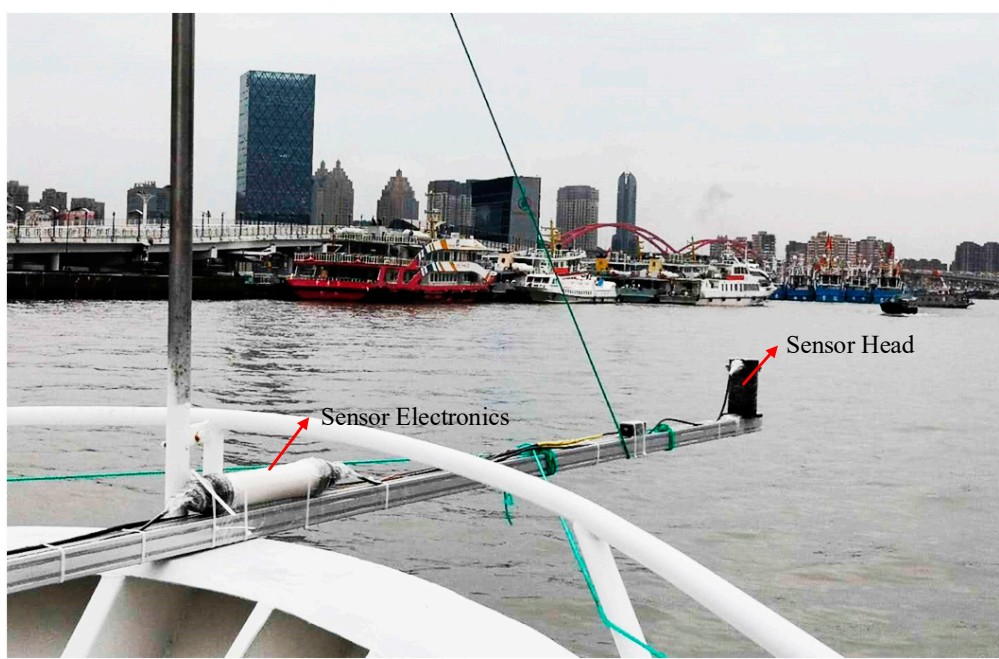

**Figure 12.** Equipment installation diagram.

Given the north–south orientation of the subsea cables, the vessel traversed east–west to cross them. During this voyage, there was a substantial variation in longitude, while latitude changes remained minor. As a result, Figures 10 and 13 exclusively showcase the correlation between navigational longitude data and the magnetic field curve. The magnetic field is an AC signal due to the 50 Hz current carried in the cable, and the amplitude of the magnetic field curve was plotted.

There were a total of four sets of subsea cable magnetic field data available for analysis. The first set of them is shown in Figure 10. The nine subsea power cables could be tracked, and nine points were located, which can be found in Figure 14. Notably, a strong alignment was observed between this magnetic field data set and the simulation results depicted in Figure 9. The presence of two peaks situated between cable D and cable F, with maximum magnetic field values, is especially prominent. Furthermore, the non-monotonic magnetic field variation between cable C and cable D is attributed to the greater distance between them. According to the results of the simulation and experiment, the magnetic field of one cable is affected by other cables, but the positioning of the cable based on the valley value of the magnetic field is not affected.

The second set of subsea cable magnetic field data is presented in Figure 13a. Notably, the entire curve demonstrates symmetry. This symmetry indicates that the ship crossed cables D to I twice, resulting in the localization of two points on each of these cables. Similarly, the other three sets of subsea cable magnetic field data successfully led to the localization of 42 points of subsea power cables. The precise positions of each of these points are identified in Figure 14.

By conducting experiments in the ocean, 51 points of subsea power cables were located. The purpose of this experiment was to detect cables G–I. The original laying GPS data of cables G–I were known, but not those for other cables. The located points and the actual

paths of the cables are shown in Figure 14. Only one point was located on cable A. The routes of cables B–I were all tracked successfully. Since the original laying GPS data of only three cables were known, we analyzed the positioning errors of these three cables. The maximum positioning error was 7.72 m at the H-6 position, and the minimum positioning error was 0.78 m at the G-2 position. This error may be caused by the fact that cables were not laid in the designated locations and their positions will have changed due to ocean currents over time. In conclusion, this series of experiments underscores the feasibility of route tracking for subsea power cables. While challenges like positional variability exist, the successful localization of multiple points along these cables showcases the potential and promise of this approach.

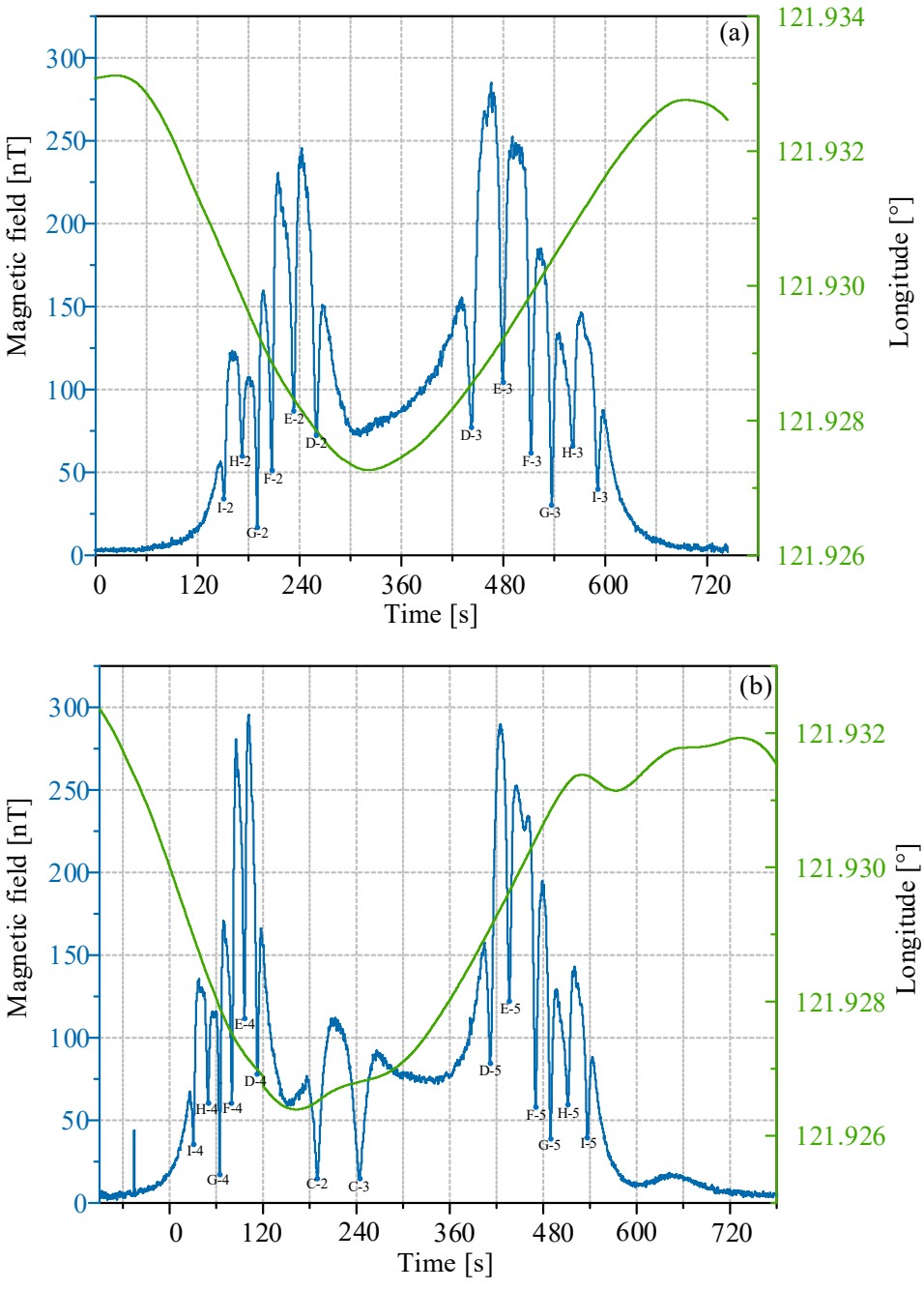

**Figure 13.** *Cont.*

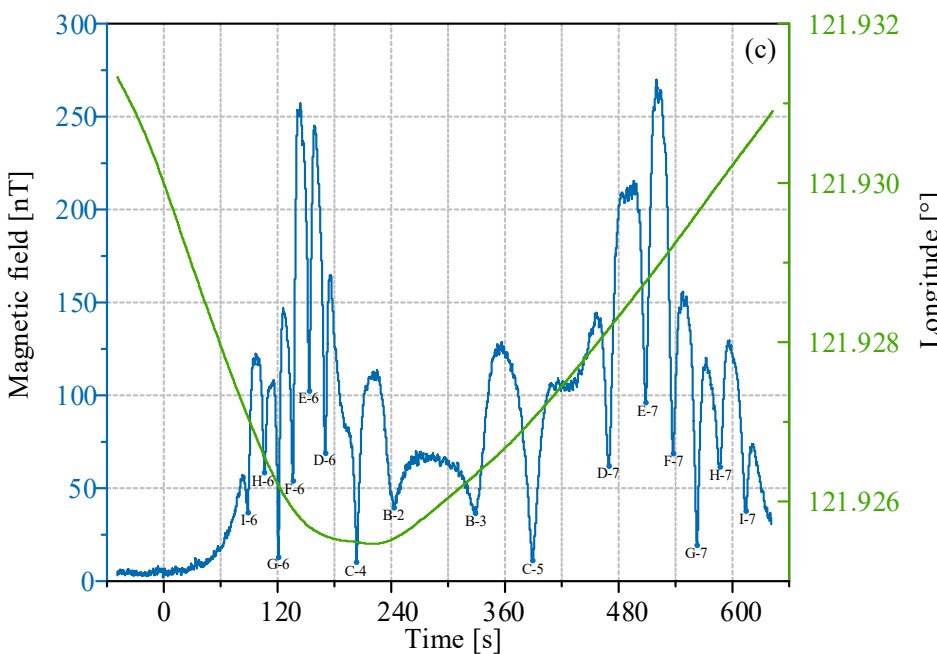

**Figure 13.** Three sets of subsea cable magnetic field data: (**a**) 12 positioning points of 6 cables, (**b**) 14 positioning points of 7 cables, and (**c**) 16 positioning points of 8 cables. Green curve: navigation longitude; blue curve: magnetic field data.

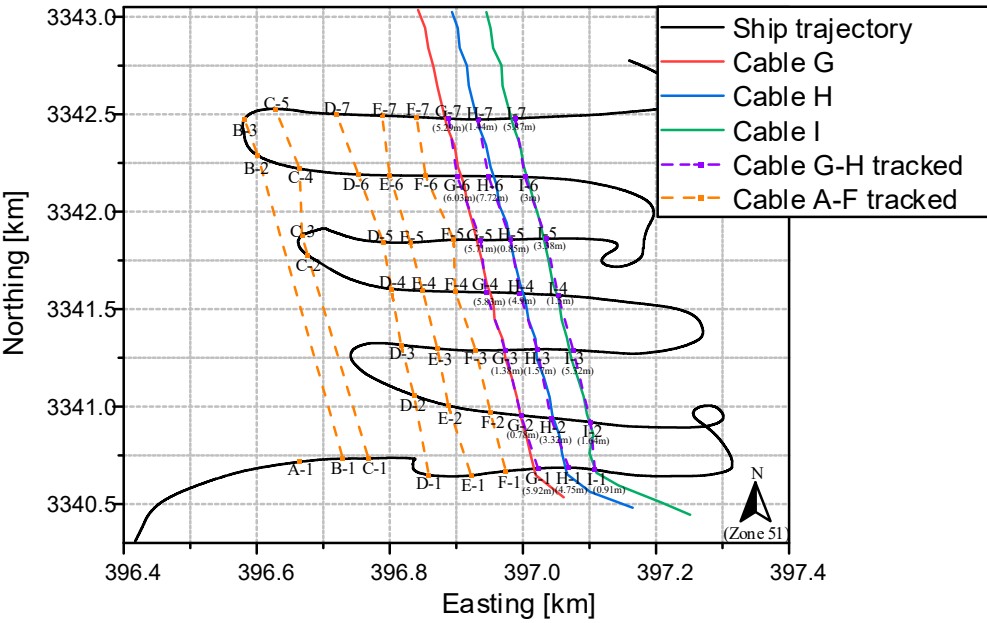

**Figure 14.** Subsea power cable route tracing results, with positioning errors within parentheses.

## 6. Conclusions

In this paper, an attitude-independent route tracking method with a self-oscillation OPM for subsea power cables, which is achieved by using scalar checking, was proposed. The excellent performance of the OPM enables the possibility of subsea cable positioning in high sea conditions. Additionally, the response of the OPM is extremely fast, so the signal of the power frequency magnetic field can be detected without any appreciable lag in response, achieving real-time capability.

Through theoretical analysis, the attitude-independent performance of the OPM and the subsea cable positioning method using scalar checking were explained in detail. Sub-

sequently, a laboratory experiment was undertaken to validate the attitude-independent performance of the OPM. To further support the sea experiment, a simulation was conducted to acquire magnetic field distribution data concerning multiple cables.

In conclusion, the result of the sea experiment confirmed the feasibility of route tracking based on the OPM using scalar checking. Remarkably, the result matched the simulation perfectly, affirming the robustness of the proposed method. Leveraging the advantages of the OPM, this approach proves highly adaptable even in challenging high sea conditions. This paper provides clear insights into the crucial role of the OPM utilizing scalar checking in subsea cable detection. Importantly, the inherent advantages of the OPM make it particularly well-suited for application in high sea conditions. By emphasizing the significance of the OPM and scalar checking in subsea cable detection, this paper effectively underscores the method's substantial potential contribution to this domain.

**Author Contributions:** Conceptualization, S.L. and G.Y.; Data curation, G.L. (Guozhu Li) and X.G.; Formal analysis, X.G. and G.H.; Funding acquisition, G.L. (Gaoxiang Li); Investigation, Y.C. and X.Z.; Methodology, G.H. and G.L. (Gaoxiang Li); Project administration, X.Z. and G.Y.; Resources, S.L. and X.Z.; Software, G.L. (Guozhu Li) and Y.C.; Supervision, G.H. and X.Z.; Validation, G.H. and G.Y.; Visualization, G.L. (Guozhu Li) and G.L. (Gaoxiang Li); Writing—original draft, G.L. (Guozhu Li) and X.G.; Writing—review and editing, G.H. All authors have read and agreed to the published version of the manuscript.

**Funding:** This research was funded by the National Natural Science Foundation of China, grant numbers 12174139 and 12374330.

**Data Availability Statement:** The data are available from the corresponding authors upon reasonable request. The data are not publicly available due to they are part of ongoing research or development, and premature release could undermine the goals of the project.

**Conflicts of Interest:** The authors declare no conflict of interest.

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
