# Peer review of "Attitude-Independent Route Tracking for Subsea Power Cables Using a Scalar Magnetometer under High Sea Conditions"

_remotesensing, doi:10.3390/rs16020226_

Round 1

Reviewer 1 Report (Previous Reviewer 3)

Comments and Suggestions for Authors

I recommend this revised version for publicationl

Author Response

Thank you for reviewing my manuscript.  I appreciate your time and consideration.

Reviewer 2 Report (Previous Reviewer 1)

Comments and Suggestions for Authors

The paper is improved and it can be accepted.

Author Response

Thank you for reviewing my manuscript.  I appreciate your time and consideration.

Reviewer 3 Report (New Reviewer)

Comments and Suggestions for Authors

In this manuscript, the authors proposed a attitude-independent route tracking method with an OPM for subsea power cables. The whole work presented in this manuscript is quite solid and interesting. However, some suggestions are presented as follows.

1. The introduction part is quite poor. The authors are suggested to identify the main published works' drawback in terms of assumptions, algorithm and others aspect. The current version of this manuscript looks mass without any clear flow.

2. As authors mentioned "The disadvantage of these magnetic field sensors is that the detection results will be affected by the determination of the magnetometer bias vector, scale factors and non-orthogonality corrections". This summary can be declared  at the beginning of this manuscript to give reader a full picture before going through all content.

3. In section 2, the authors introduce "Attitude-independence for Self-Oscillation OPM". If just an introduction, this part can be moved to introduction. If not, what is the main contribution of Section 2?

4. Eq.(6) shows the Taylor expansion. How is it used in practice? Any approximation?

5. Eq(20) can be introduced briefly if possible.

6.  All symbols and notation should be clearly presented their physical meaning before using them.

Comments on the Quality of English Language

The authors should be carefully revised wording of this manuscript. 

Author Response

Reviewer 4 Report (New Reviewer)

Comments and Suggestions for Authors

The paper with title "Attitude-independent Route Tracking for Subsea Power Cables using Scalar Magnetometer under High Sea Condition" presents a method for tracking subsea power cables in real-time, irrespective of the platform's attitude. It uses a high bandwidth self-oscillating optically pumped magnetometer to measure the magnetic field produced by the current in a cable, which is immune to platform attitude changes. The approach is theoretically and experimentally investigated, demonstrating its effectiveness in high sea conditions.

Τhe strong points are:

Innovative Methodology: The use of a self-oscillating optically pumped magnetometer for attitude-independent detection is a novel approach in this field.

Comprehensive Experimental Validation: The paper includes both laboratory and sea experiments to validate the theoretical model.

Real-world Application: The method addresses a significant challenge in subsea cable tracking, especially under high sea conditions, which is critical for the maintenance and monitoring of these infrastructures.

The weaknesses points are:

Complexity of Implementation: The technical complexity of the proposed method might limit its practical applicability in certain scenarios.

Dependence on Specific Equipment: The method relies heavily on a specific type of magnetometer, which may not be widely available or could be costly. Are there other alternative devices that could be used?

Potential Environmental Limitations: While the method is tested under high sea conditions, its effectiveness in different types of marine environments (varying salinity, temperature, etc.) is not clearly addressed.

Lack of Comparative Analysis: The paper could be strengthened by comparing the proposed method with existing techniques to highlight its advantages and potential limitations in a broader context.

Overall, the paper contributes significantly to the field of marine technology and subsea cable maintenance, offering an innovative solution to a challenging problem.

Author Response

Thank you for reviewing my manuscript.  I appreciate your time and consideration.

Please review the attachment for responses to the weaknesses points.

This manuscript is a resubmission of an earlier submission. The following is a list of the peer review reports and author responses from that submission.

Round 1

Reviewer 1 Report

Comments and Suggestions for Authors

1. Please show more information about the properties of the optically pumped magnetometer, such as the magnetic field resolution, dynamic range and the bandwidth.

 2. How about the amplitude of the current in the power cable?

3. How about the speed of the boat?

Reviewer 2 Report

Comments and Suggestions for Authors

This is good investigation for subsea cable detecting based on measurement of the magnetic field. Simulation and laboratory experiments have been done. Further, sea experiment has been done, in which cables in subsea, marked from A to I, were situated at an approximate depth of 20 m. In most cases, many cables are located at deeper seafloor, such as 1000-2000 m. So, the problems are: Is this method still valid in these cases? How to apply this method in these cases? such as by using AUV to take magnetic sensors. I think authors should consider these problems carefully.

Thank you very much!

Comments on the Quality of English Language

No problem in English!

Reviewer 3 Report

Comments and Suggestions for Authors

This is a well-written manuscript which I recommend for publication.  The use of the word "attitude" in the title, abstract and text is new to me.  Perhaps the authors might provide a sentence of clarificationl

Reviewer 4 Report

Comments and Suggestions for Authors

The manuscript 2610352 submitted to MDPI-RemoteSensing and entitled “Attitude-independent route tracking for subsea power cables using scalar magnetometer under high sea condition” by Guozhu Li et al. presents a method to localize/characterize undersea power cable by means of a magnetometric approach. The Authors consider the case of an optical magnetometer, stressing the advantages related to the scalar response of such a kind of sensor.

The application scope of the presented research is clear and clearly presented, but many details remain unclear or are not considered at all, which makes the main focus of the paper inadequately (and in some cases incorrectly) analyzed.

In summary, I had a generally negative impression both from the contents of this work and from the clarity of presentation, to an extent leading me to conclude that the manuscript is not worth of being further considered for publication.

I will report here below a non-exahustive list of issues to support my negative evaluation.

There exist a variety of optical atomic magnetometers (OPMs) characterized by diverse working principles and operation modes: presenting the “self-oscillating OPM” as a unique kind of sensor is misleading, and the issue is made heavier by the lack of information about the specific setup used.

Indeed no details are provided about the sensors and only qualitative information is provided about its specification.

It is true that the OPM generally produce a scalar response, but –particularly when operated in the considered conditions (Earth field, i.e. several tens of microtesla), the ideal isotropic response is lost and important issue emerge, the so-called heading-errors. There is abundant literature about this topic that should be not only cited, but also analyzed. OPM are high-performance sensors, but are definitely more complex than to be operated with in comparison with other devices such as Hall, magnetoresistive or fluxgate sensors. These latter have a vectorial response so to provide richer information for localization tasks.

It is true that vectorial sensors suffer from inhomogeneous gain or misalignments among the three axes, but literature reports extensively about methodologies to recover a good level of isotropy in gain. Authors should evaluate if the isotropy degree eventually achievable with those sensors does not make them even better that the OPM affected by non-ideal (for uncompensated heading error, i.e.  for anisotropic) response.

As correctly stated in page 5, in a first-order approximation, a scalar sensor detects small field variations only along the bias (ambient field). Apart from considering the dynamical response (I will come back later on this subject) the eq.7 express this response, which is actually vectorial because related only to the component along B_g. Thus a crucial question emerges: to which extent is the direction of B_g static? Did Author considered that the presence of ferromagnetic materials can make the environmental field not-homogeneous (and to which extent)? In addition, if the B_g direction is not identified (and this would require a vectorial sensor) the measured quantity is not completely characterized (see line 162).

The qualitative statement in line 97 (extremely fast) is not generally true: the response of these devices is not always with a large bandwidth, thus qualitative assessments about the B_c strength can be wrong if the dynamical response of the magnetometer is not taken into account. There is abundant literature also about this subject.

I am now continuing with a set of secondary remarks, but also this list will not be exhaustive.

Title:  Attitude with the meaning of orientation could be a jargon derived from navigation, I would suggest the use of less specialistic terms, such as “orientation”. However, as a non native speaker, I could be wrong in this respect.

L25 is
àare?
L32, 33, 50 AUV and ROV are undefined
L66 what is it intended for multi-probe? Is it a device with more than one magnetometric sensor, or an arrangement of different kind of sensors?

L67 information about those commercial devices (working principle, specification, references) should be provided

L69 I assume that the necessary calibration procedures are developed, characterized and assessed: do they constitute severe limitations?
L77, 83 typos

L92-95 the description of the OPM working principle is extremely rough and inappropriate

It is correct to state that the OPM are extremely sensitive: they achieve state-of-the-art performance in terms of sensitivity. But which sensitivity level is really needed in the considered application? Quantitative analyses of the available technologies would be required and limiting the discussion only to the scalar/vectorial response is -to my opinion- misleading.

L95 magnetometers or magnetometer?

L101-104 the sentence needs to be reformulated

Page 16 the description of the OPM is inaccurate and not clear, both in term of Physics and of English

L120 not understandable: what is the quantity being substituted?

L 117, 125 check English

In eq.6, x is not defined

L170 on which direction/plane is that field projected?

L172, 190 define inclination and dip: with respect to what?

L189 check (alpha or its sine? Is it fully equivalent)

L218 why 27 Hz and not closer (and possibly above) the target frequency of 50Hz?

L243 why does multi cable require numerical tools? Isn’t the eq.3 and the superposition principle sufficient to estimate the field of a set of straight cables?

L266 is each of the 9 cable a single wire or a dual-three phase cable?

L295 what does it mean that the field of one cable is affected by other cables? Is the superposition principle not valid? I assume the measured quantity is the sum of different contributions, but those contribution do not interfere with each other. Am I wrong?

Figs 11 and 12: why not showing also a field versus position plot (similar to fig.9)?

Is the (nearly parallel) orientation of the cables relevant? Could they be more variously oriented or even, possibly crossing ?

Fig.13 a colour map would probably help the reader to better visualize the field strength as a function of the vessel position (longitude and latitude)

There are references from the same authors with pretty similar titles (e.g about works presented as  proceedings and as regular papers): are they all worth of being cited?

I did not find references about the specific self-oscillating OPM (its structure and its specifications)

The task of magnetometrically identify specific magnetic sources is not limited to the considered application: both static and alternating field sources are localized/tracked in a variety of concurrent application (from robotics to medicine and security): I assume that literature is missing in this respect. If so, the introduction could be correspondingly  incomplete.

In ref 36, the first name is wrongly spelled (it should be Di Domenico, G.)

Comments on the Quality of English Language

I think that several sentences require a deep revision, because I could not understand their meaning (see the main report). However, as a non-native reader, I cannot provide a more accurate evaluation about smoothness and precise wording of the text.

Round 2

Reviewer 4 Report

Comments and Suggestions for Authors

I have been invited to provide a second-round revision of the manuscript 2610352 submitted to MDPI-Remote Sensing and entitled “Attitude-independent route tracking for subsea power cables using scalar magnetometer under high sea condition” by Guozhu Li et al. In my first-round report I expressed a generally  negative evaluation, suggesting the Editor to reject the manuscript and the Authors to resubmit a completely refreshed version, if they were interested in publishing their experimental data.

In this occasion, I realize that three out of four reviewers got amazingly different conclusions, suggesting an immediate acceptance or asking for minor revisions. I do not know neither if the other peers will have the opportunity of reading and evaluating my own remarks, nor if they are requested to evaluate their appropriateness and, in case, to examine if the Authors have adequately addressed them.

It can be easily argued that my opinion in this occurrence is likely biased by the deepness of my previous objections, and being my evaluation the only negative one among four, it is probable that the manuscript will be finally accepted.

I wish just to underline persisting serious weaknesses of the manuscript, mentioning a few of them, among the most significant.

The answer to my comment n.3 is not correct when saying “The primary ferromagnetic material is the body of the ship, which can be considered as a dipole. Its magnetic influence on the geomagnetic field is constant and does not change with alterations in the heading.” The mistake can be explained as follows: the field generated by moving ferromagnetic materials (the ship) sums vectorially to the earth field. If the term generated by the ship is not negligible with respect to B_geo, the resulting static field will be differently oriented with respect to B_geo, the fact that the summed quantities are vectors causes that, when one of the to terms change orientation with respect to the other (i.e. when the ship rotates around any axis) the total field will change orientation: the sum of two vectors of given intensity produces a vector that changes when the relative direction of the two addends changes.

The statement that a large ferromagnetic source produces a dipolar field is wrong unless the field is considered at a large distance from the source –i.e. the ship- size (I do not think that this is the case, as can also be seen in fig.10). The intensity could be small regardless of the dipolar nature of the field, but this requires a small average magnetization of the source: an aspect that seems to have been overlooked.

The response to my comment n.22 is clear but partial. The final quantities to be measured are at 50Hz, a frequency at which the mains produce technical noise both of magnetic and of electric nature. It can be argued that this problem is crucial in a building-based lab and relaxed on a vehicle if no 50Hz supplies are around. Is it sure that no 50Hz technical noise will affect the in-field measurements, as to make the 27Hz simulation in the lab really significant?

I could not understand the response to my comment n.23: does the superposition principle fail when many cables are considered? Or, why the complexity of finite element simulation becomes lower than calculation with superposition principle at increasing number of cables?

The statement which links the instrumental bandwidth to the counter throughput rate (line 100) is most probably wrong. The reason is that a counter, with a measurement of a finite duration has an intrinsic uncertainty in frequency (something related to the Fourier indetermination principle). Converting such a frequency uncertainty to field uncertainty using the gyromagnetic factor (e.g. 3.5Hz/nT for Cesium) would lead to 100nT levels: is it so? In this case, what the 3pT mentioned in line 99 refers to? More generally, as I tried to point out in my remarks n.4, the dynamical response of the magnetometer must be taken into account, and a direct conversion from static calibration is not feasible.
I think that the reported 3pT resolution is indeed a 3pT/sqrt(Hz) sensitivity (see ref 30 page viii), meaning that at 250Sa/s the resolution is correspondingly lower, apart from the mentioned uncertatinty due to using a counter.

I guess that the real measurement lasts several periods of the ac field under measurement, which would enable the application of the FFT algorithms mentioned at the line 233 (and I must say that more advanced tools are available to identify harmonic signals, see e.g. T.Grandke IEEE 83: 10.1109/TIM.1983.4315077 )

The theoretical description of the OPM does not seem to be particularly relevant to describe this systems and leaves open questions about its dynamical response.

The sentence “To model the magnetic field interactions of multiple cables” similarly to what I wrote in my remark n.25 seems a non-sense: superposition is not interaction. The fields generated by diverse sources do sum to each other (linearly and vectorially), but do not *interact* with each other.

Apologize for the typo that occurred in my comment 6 making it not understandable: it should have been “IS or ARE?”